# Fidgety Movements Assessment Accuracy Survey in Japan

**DOI:** 10.3390/ijerph182413428

**Published:** 2021-12-20

**Authors:** Tomoki Maeda, Osamu Kobayashi, Kenji Ihara, Arend Frederik Bos

**Affiliations:** 1Department of Pediatrics, Oita University Faculty of Medicine, Oita 879-5593, Japan; o-kobayashi@oita-u.ac.jp (O.K.); k-ihara@oita-u.ac.jp (K.I.); 2Division of Neonatology, Beatrix Children’s Hospital, University Medical Center Groningen, University of Groningen, 9713 GZ Groningen, The Netherlands; a.f.bos@umcg.nl

**Keywords:** general movements, fidgety movements, self-learning exercise

## Abstract

To investigate the accuracy of fidgety movements (FMs) assessment in Japanese assessors. Sixty specialists participated in the first survey. Of the participants, 18 were assessors certified by the GMs basic-training course. The surveys were composed of FMs assessment of 20 video clips. The correct assessment rates (CARs) were investigated. The survey videos were judged into three types: normal (F + +, F +), abnormal (AF) and absent (F + -, F -). After the first survey, each participant performed a self-learning exercise using clips of the first survey. The follow-up survey was conducted three months after the first survey. The median CAR of the first survey was 65% in certified assessors and 50% in noncertified assessors. The median CARs of certified assessors were significantly higher than that of noncertified assessors for all clips and for normal FMs and AF clips (*p* < 0.01). After 3 months of self-learning exercise the CARs in each judgement type improved in 28 follow-up survey participants. Their median CAR improved from 60% in the first survey to 84% in the follow-up survey. To practise general movements assessment (GMA), course certification is required. The self-learning exercise with the confirmed judgement FMs clips may be effective for improving the ability of FMs judgement.

## 1. Introduction

General movements (GMs) are endogenously generated movement patterns in foetuses and young infants. The quality of GMs reflects the integrity of the young nervous system and serves as a predictor for later neurological outcomes [1,2]. Prechtl’s method of general movement assessment (GMA) is a method of estimating developing brain function based on video observation. GMA does not require any special equipment other than video recording and can be performed in any country, but since GMA is performed using visual gestalt perception, training is required to practice GMA [2]. The usefulness of GMA for predicting motor dysfunction, especially cerebral palsy (CP), has been established. According to a systematic review of the best available evidence for the early accurate diagnosis of CP, GMA has been strongly recommended for the early detection of CP [3]. GMA has been increasingly used in many countries. 

Japan is a country with excellent perinatal care, where the neonatal mortality rate is less than 1 per 1000 live births. On the other hand, the prevalence of CP per 1000 population in Japan is reported to be 2.27 [4], which is higher than that of other developed countries [5]. The prevalence of CP might be high because many extremely preterm infants have survived. Although GMA is recommended for early detection and intervention for CP, it is not widely practised in Japan. This is because there are few GMA training courses and only few assessors are certified regarding reliability of GMA. Understanding the current status of GMA accuracy in Japan is extremely important when considering how to practise GMA there widely. Therefore, the purpose of this study was to investigate the accuracy of fidgety movements (FMs) judgements in Japanese assessors. 

GMs occur in age-specific patterns. Before term age they are called foetal and preterm GMs, around term age writhing movements and from 9 weeks post-term age onwards FMs [1,2]. We decided to assess only FMs in the present study for three reasons. First, FMs are more useful in predicting CP than preterm and writhing GMs [6]. Second, FMs are frequently observed in awake and active alert states, so they have the advantage of requiring less time for video recording and assessment [2]. Third, the inter- and intra-observer agreement of GMA have been reported to be higher in FMs than in writhing GMs [7]. 

In Japan, GMA training courses have been held only once. The basic training course was held in October 2018, had 28 Japanese participants, and they were certified through the course. Only a few assessors have taken GMA courses abroad. As a result, the current situation in Japan is that GMA accuracy is not guaranteed, and practise is limited. A small number of certified assessors communicated with each other and have striven to maintain the accuracy of GMA. Recently, to make matters worse, the opportunity for face-to-face discussions between assessors has been severely restricted due to the COVID-19 pandemic. These bad situations have a serious impact on maintaining GMA skills appropriately. To countermeasure this serious issue, we conducted a survey on the current status of GMA accuracy in Japan. Furthermore, we present a trial of self-learning judgment exercises at home. This paper should not only be helpful to Japan, but also to assessors worldwide, who currently have very limited opportunities to discuss GMA.

## 2. Materials and Methods

### 2.1. Survey Participants

The first survey was conducted from May to June 2020. Participants were recruited through the Japanese GMs practise network built on the only GMs basic-training course, which was held in Japan in October 2018. Anyone who is interested in GMs can register with the network, not just those who participated in the basic-training course. The participation requirement for this survey was a medical/rehabilitation specialist who knew basic knowledge of GMs. The basic knowledge to participate was to understand the difference between abnormal fidgety (AF) and absent fidgety (F-). 

Along with the survey, we asked three questions that could be answered “yes” or “no” about the GMA experience. (1) Do you have been certified the GMs training course (2) Do you practise GMA regularly for clinical or research purposes? (3) Have you watched more than 3 videos of fidgety GMs in the last 3 months?

After explaining that submitting the scoring sheet would be regarded as consent to participate in this study, the participants voluntarily participated in these surveys. This investigation was approved by the Ethics Committee of Oita University.

### 2.2. The Temporal Organization of FMs

FMs are small movements of small amplitude and moderate speed and variable acceleration of the neck, trunk, and limbs in all directions [2]. FMs usually occur among infants aged between approximately 9 weeks and 20 weeks post- term age. The temporal organization of FMs can be defined as follows [8]: Continual FMs (F + +); Continual FMs frequently occur in the whole body, though interspersed with very short (1–2 s) pauses. Intermittent FMs (F +): Intermittent FMs occur in all body parts, though with longer pauses (up to 10 s). Sporadic FMs (F + -): Isolated fidgety bursts of 1 s to 3 s are interspersed with long pauses of up to 1 min. Absent FMs (F-): FMs are missing from 9 to 20 weeks post-term age. Abnormal FMs (AF): Abnormal FMs look like normal FMs, though with a greater amplitude, speed, and jerkiness. F + + and F + are normal, and F + -, F -, and AF are abnormal for this period. In the survey, the videos were judged into three types according to the temporal organization of FMs. The three types are normal (F + +, F +), abnormal (AF) and absent (F + -, F -). Table 1 shows the definition of the judgement type and the temporal organization of FMs. 

### 2.3. Confirmed Video Clips for the Surveys

The video clips assessed in the surveys were constructed from video clips recorded during GMs clinical research conducted at Oita University. The clinical research was approved by the Ethics Committee of Oita University. The use of video clips for this survey was included in the written parental consent to participate in the clinical research. The video clips of preterm, term infants, cases with brain lesions such as intraventricular haemorrhage, and healthy infants were included, which were recorded at 9 to 17 post-term weeks. All video clips were reviewed and confirmed the judgement by the author TM (advanced course certified assessor) and AFB (GM trust tutor).

The number of video clips to assess was 20, and each clip was edited to 1 min. The numbers of clips by judgement type were 15, 2, and 3 clips for normal (F + +, F +), AF, and absent (F + -, F -), respectively (Table 2). The clips were deleted from the audio information and processed mosaicking on the face of the infant. The video clips were recorded following the GMs standard recording methodology such as the position of the infant lying in the supine position and wearing only diapers.

We present the examples of the survey video clips in Appendix A. The survey videos were one minute long, with mosaicking on the face. The judgement types of clip E, F and P are F + +, AF and F + -, respectively (Appendix A).

### 2.4. Survey Implementation Method

Before the survey, the survey organizer provided participants with a learning digital versatile disc (DVD) which demonstrated FMs different from the survey video clips. The DVD consisted of demonstration video clips of 3 normal judgements (2 clips F + + and 1 clip F +) and 3 abnormal judgements (1 clip AF, 1 clip F + -, and 1 clip F -). Participants in the survey were free to learn on DVD before undergoing the survey.

The Survey DVD containing the assessment video clips was given to the survey participants, and 20 video clips were assessed within one consecutive hour. The 20 clips named A through T were located on the DVD in alphabetical order. Table 2 shows the details of the clips. There was only a total assessment time limit. The playback order and number of playbacks of each video clip were left to the participants. Participants were informed that the video recordings were carried out at 9–18 post-term weeks, but the exact post term week at recording and birth gestational age of the assessing case were not provided at the time of assessment.

Following the definition of Prechtl’s GMA the video clips were assessed into 3 judgement types: normal FMs (F +, F + +), abnormal FMs (AF), and absent FMs (F + -, F -) [2]. The surveys were conducted without informing participants how many of each judgement type were included in the 20 video clips.

Because this short GMA survey mimics a situation that applies to identifying cases that require detailed assessment, the survey organizer (TM) instructed participants to assess the judgement as abnormal if it was unclear whether the judgement of the video clip was normal or abnormal.

### 2.5. Self-Learning Exercise after the Survey

After the survey, the survey organizer held a web meeting on the assessment with participants. The web meeting consisted of providing and explaining the correct answer and discussion on the findings of the survey video clips. After that, participants continued self-learning exercise using the DVD provided for the survey.

### 2.6. Follow-Up Survey

The follow-up survey was conducted in August 2020 three months after the first. We recruited follow-up survey participants from the first survey. The participation requirement was that the assessor thought he or she could assess correctly when the same video was shown as during the first survey. 

The follow-up survey was conducted in the same manner as the first. The video clips now consisted of 20 clips named A2 through T2. Five of these were created by flipping the first survey clips horizontally. The CAR of the follow-up survey was evaluated by the 15 new non-flipped videos, excluding the flipped clips. Table 2 shows the details of the clips.

### 2.7. Data Analysis

The concordance rate of the judgements in this survey with respect to the confirmed GMs judgements was calculated as the correct assessment rate (CAR) and used for analysis. Since the normal distribution of CARs in abnormal and absent judgement types could not be confirmed, we presented the results by the median (inter quartile range; IQR). The data were analysed using SPSS Statistics version 24 (IBM, Tokyo, Japan). We statistically examined CARs using the Mann–Whitney U test. Differences with *p* < 0.01 were considered significant. The survey judgement sheets were anonymized and collected. Therefore, the individual judgement sheet of the first survey and the follow-up survey cannot be linked. Since we can recognize the sheets from facilities that participated in both surveys, we compared the sheets from the facilities that participated both surveys to evaluate the effectiveness of self-learning exercise.

## 3. Results

### 3.1. Participants

Sixty specialists participated in the first survey from nationwide institutions in Japan. Participants included 11 medical doctors, 41 physiotherapists, and 8 occupational therapists. Of the participants, 18 participants were assessors certified by the GMs basic-training course, which is approximately half of all certified assessors in Japan. Forty-two participants were non-certified assessors. Twenty-one participants reported that they were practising GMA regularly. Seventeen participants had assessed GMs in 3 or more video clips during the preceding 3 months (Table 3). Twenty-eight participants participated in the follow-up survey (Table 4).

### 3.2. The Accuracy of FMs Judgements in Japanese Assessors

The video clips for the survey consisted of 20 clips named A through T. Table 2 shows the CAR for each clip. The median CAR of the first survey was 65% in certified assessors and 50% in noncertified assessors. The median CARs by judgement type were normal FMs 67%, AF 50%, and absent FMs 67% for certified assessors and normal FMs 53%, AF 50%, and absent FMs 33% for noncertified assessors. Of the normal FMs, the CARs for F + + clips were 90% among certified assessors and 70% among noncertified assessors. The median CARs of the certified assessors were significantly higher than that of noncertified assessors for all clips and for normal FMs, F + + clips and absent FMs clips (*p* < 0.01) (Figure 1). There was no difference in the CARs depending on whether practising GMA regularly, and whether having watched 3 fidgety GMs videos in the last 3 months. (Table 3).

### 3.3. The Follow-Up Survey

The video clips for the survey consisted of 20 clips named A2 through T2. Table 2 shows the CAR for each clip. The median CAR of the follow-up survey was 84%. The median CARs by judgement type were normal FMs 82%, AF 100%, and absent FMs 100%. Among the normal FMs, the CARs for F + + clips were 82%. These CARs were from 15 clips excluding the flipped clips. 

Of the 28 participants in the follow-up survey, 12 assessed all flipped clips correctly, and 10 assessed 4 clips correctly, with a total of 22 assessors achieving a CAR of 80% or more on flipped clips. In order to examine the CARs whether self-learning exercise is effective, we evaluated the CARs in these 22 assessors by only non-flipped video clips of the follow-up survey. The median CAR of non-flipped video clips in the follow-up survey was 87% for those who had a CAR of 80% or more on the flipped clips (Table 4).

### 3.4. Comparison of the First and the Follow-Up Surveys

We compared the CARs of the first and the follow-up survey in the 28 follow-up survey participants. Their median CARs were 60% in the first survey, and 84% in the follow-up survey. The median CARs by judgement type were normal FMs 67%, AF 50%, and absent FMs 33%. Among the normal FMs, the CARs for F + + clips were 67% in the first survey. The median CARs were normal FMs 82%, AF 100%, and absent FMs 100%. Among the normal FMs, the CARs for F + + clips were 89% in the follow-up survey. The median CARs showed significant increase in the normal FMs and absent FMs (*p* < 0.01) (Table 4) (Figure 2).

## 4. Discussion

In the present study, the median CAR of the first survey was 65% for certified assessors. The certified assessors had significantly higher CARs than the non-certified assessors in all judgment types except AF. There were no differences in the CARs between participants who practiced GMA regularly and those who did not, nor between participants who watched three fidgety GMs videos during the preceding 3 months and those who did not. These results clearly indicated that practising GMA requires the course certification. It is difficult to practise FMs assessment only by learning from books and /or demonstration videos. In Japan, the medical and rehabilitation specialists cannot master the FMs assessment skills because even if they try to practise GMA, only few assessors are able to maintain assessment skills. This means that there are few opportunities for the assessors to confirm the accuracy of their own GMA results.

The CAR of the F + + judgement type was 90% but the CARs of the other judgement type were so low that it is insufficient for using GMA in clinical practise, even for certified assessors. Valentin et al. reported that the ability to assess GMs correctly can be gained from just receiving a standardized training course of a few days [9]. The authors reported that the CAR for FMs was 87% at the end of the training course. The certified assessors who participated in the present study were confirmed the accuracy of the assessment at the end of the basic-training course which was held in 2018 in Japan. The results of the present study demonstrate that their assessment ability declined during the one and a half years after completing the basic training course. Although F + + can be recognized, the assessment accuracy of cases that require judgement of F + -, F - and AF had declined. The accuracy for suspicious abnormal judgement types is the most important in clinical practise. The countermeasures are required to maintain the assessment ability in certified assessors.

Examining the individual video clips, the CARs for the first survey in certified assessors for clip B in the F + + type, clips H and clip O in the F + were very low at 17%, 6%, and 6%, respectively. Clip B had many FMs accompanied with other large, fast movement patterns such as swipes and kicking, and is suspicious for being F ++ or AF. Clip H and O had FMs but they were not continuous appearance, and are suspicious for being F +-. All of these were the video clips that the assessment judged as normal but were suspected abnormal patterns. This is because this survey judgement was intended to be applied for identifying cases that required detailed assessments. Therefore, the assessor was instructed to assess as an abnormal judgement when he/she was worried about whether the FMs were normal or abnormal. Even if the assessment bias in screening is taken into consideration, the CARs 65% in the first survey were low for certified assessors.

In the comparison between the first survey and the follow-up survey, the median CARs showed a significant increase in normal FMs and absent FMs. The follow-up survey was done 3 months after the first survey. Follow-up survey participants performed self-learning exercises and could learn to assess the various videos from the first survey correctly. In this investigation, effectiveness by the self-learning exercise was based on self-judgement. To consider the effectiveness of the self-learning exercise, the 20 video clips of the follow-up survey included five video clips of the first survey clip, but now flipped horizontally. Significant increases in the CARs were observed for follow-up survey participants, when excluding the flipped clips and excluding assessors whose CAR of the flipped clips was less than 80% (Table 4) (Figure 2). 

The assessors practising GMA in Japan have no ways to confirm their own judgement, so even if they have practised GMA, it may not have led to an improvement in their assessment reliability. Conversely, in the present follow-up survey, the trial of self-learning exercise was effective for improving the CARs. This favourable result is probably because the videos for the self-learning exercise confirmed their accurate judgements. The self-learning exercise with the confirmed judgement FMs clips may therefore be effective for improving the ability of FMs judgement skill.

We recognize several limitations to this study. First, we tried to assess 20 clips in an hour which mimics a situation that applies to identifying cases that require accurate assessment. Therefore, the video clips were edited to 1 min, which is a short time for a usual fidgety assessment clip. Mosaic processing on the infant’s face can affect the assessor’s perception of gestalt. For these reasons, we should be careful when comparing the results of the present study with previous reports on the accuracy of GMA. However, the follow-up survey was conducted using the same methodology as the first survey, and the effect of improving the assessment accuracy was shown. Second, the difficulty of the judgements in the first and the follow-up surveys clips may not have been similar. The difference of CARs between the surveys might result from the difficulty of judgement. However, because the composition of the judgment types of the two surveys were similar, and the CARs were increasing in all judgment types, we believe that the self-learning was effective. We will provide these survey video clips to those who wish to improve their skills by self-learning, and we hope that the follow-up examination of this self-learning exercise trial will be done by other investigators. This self-learning exercise may be effective only under special circumstances in Japan, where few certified assessors but many specialists try to practise GMA by learning GMA. Therefore, it is expected to investigate countries other than Japan in the future. Third, this survey was conducted mainly through communication on the web. Compliance with the method of the survey was due to self-reporting. Therefore, survey conditions may not be uniform among participants. To investigate the effect more accurately, it is necessary to unify the investigation conditions strictly. On the other hand, this study proved that the CAR could be increased by connecting with many assessors on the web and conducting self-learning exercises. Interactive discussion between assessors is important for maintaining GMA accuracy. Unfortunately, the opportunity for face-to-face discussions between assessors has been severely restricted due to the COVID-19 pandemic now. This paper should also be helpful to assessors around the world, who currently have very limited opportunities to discuss GMA.

## 5. Conclusions

To practise GMA requires certification following a basic GM course. Under the current status in Japan, the ability to assess the quality of FMs, especially that for abnormal patterns, was not accurate enough for use in clinical practise, even after having been certified following a completed GM course. The self-learning exercise with the confirmed judgement FMs clips might be effective for improving the ability of FMs judgement skill.

We can provide the video clips used in these surveys to researchers wishing to re-examine this survey result. Please contact the corresponding author.

## Figures and Tables

**Figure 1 ijerph-18-13428-f001:**
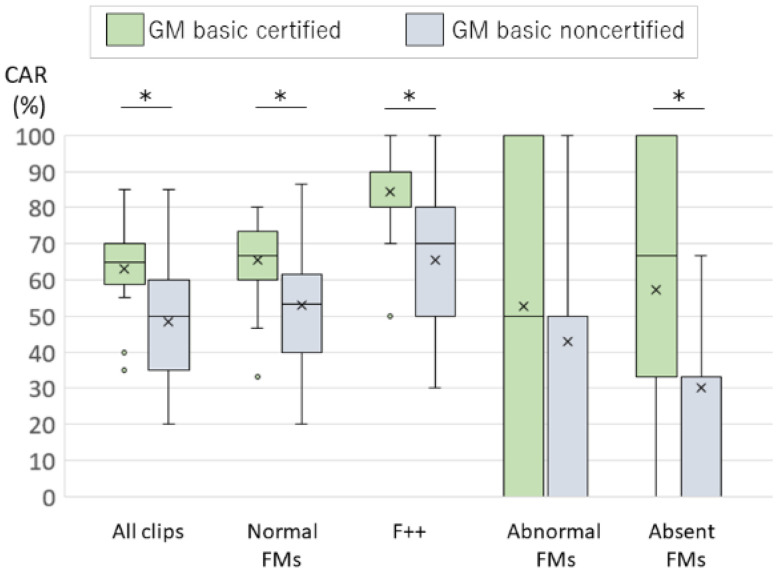
The comparison of the CARs between the GM basic course certified and noncertified assessors in the first survey. (The CARs of the certified assessors were significantly higher than that of noncertified assessors for all clips and for normal FMs, F + + clips and absent FMs type. The box plot indicates upper and lower quartiles. The whisker plot indicates the maximum and minimum of the data. Outliers are plotted as individual points. * *p* < 0.01).

**Figure 2 ijerph-18-13428-f002:**
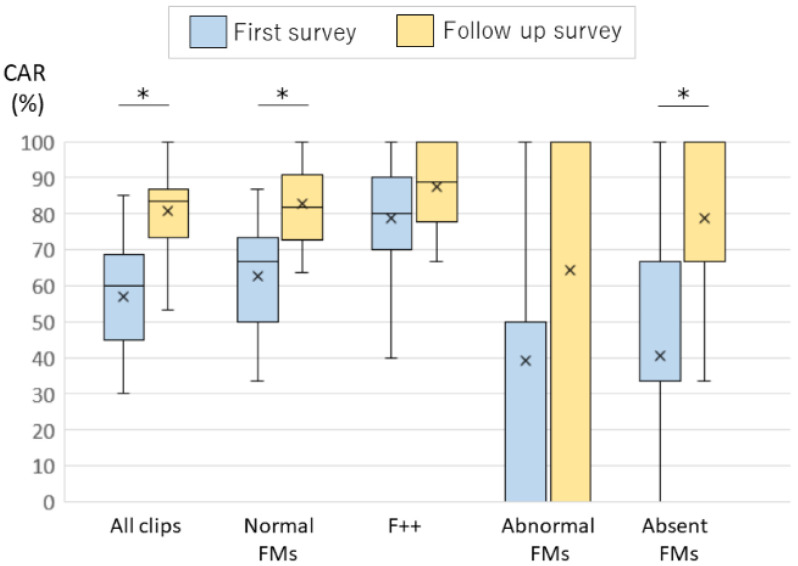
The comparison between the CARs of the first and the follow-up surveys by judgement type. (The CARs were significantly increased for all clips and for normal FMs and absent FMs type. The box plot indicates upper and lower quartiles. The whisker plot indicates the maximum and minimum of the data. * *p* < 0.01).

**Table 1 ijerph-18-13428-t001:** The definition of the temporal organization of fidgety movements.

Judgement Type	Temporal Organization		
normal	F + +	continual FMs	FMs frequently occur in the whole body, though interspersed with very short (1–2 s) pauses.
	F +	intermittent FMs	FMs occur in all body parts, though with longer pauses (up to 10 s).
abnormal	AF	abnormal FMs	Abnormal FMs look like normal FMs, though with a greater amplitude, speed, and jerkiness.
absent	F + -	sporadic FMs	Isolated fidgety bursts of 1 s to 3 s are interspersed with long pauses of up to 1 min.
	F-	absent FMs	FMs are missing altogether from 9 to 20 weeks post term age.

**Table 2 ijerph-18-13428-t002:** The survey video clip list and their respective CARs.

First Survey			CAR (%)		Follow up Survey				CAR (%)	
	Judgement Type	Record PT (Week)	GM Basic				Judgement Type	Record PT (Week)	GM Basic	
			Certified	Non-Certified					Certified	Non-Certified
A	F + +	15	83	52	A2		F + +	13	89	54
B	F + +	15	17	26	B2		F + +	13	100	100
C	F + +	14	100	98	C2		F + +	13	100	92
E	F + +	12	78	45	I2		F + +	11	89	77
L	F + +	13	94	79	J2		F + +	11	89	100
N	F + +	13	94	90	L2	Flip L	F + +	13	100	100
Q	F + +	14	94	90	M2		F + +	14	100	77
R	F + +	14	100	50	O2		F + +	12	100	100
S	F + +	13	100	81	P2		F + +	13	67	77
T	F + +	13	83	43	R2	Flip E	F + +	12	100	92
G	F +	12	39	26	T2		F + +	13	100	100
H	F +	15	6	26	D2	Flip G	F +	12	78	92
I	F +	13	39	29	E2		F +	13	44	62
K	F +	14	50	52	F2		F +	13	89	62
O	F +	15	6	7	G2	Flip F	AF	15	67	92
F	AF	15	44	29	Q2		AF	13	78	54
M	AF	14	61	57	H2		F + -	11	89	69
D	F + -	11	78	64	K2		F + -	9	100	77
J	F + -	12	28	5	N2	Flip P	F + -	12	100	85
P	F + -	12	67	21	S2		F-	17	100	62

CAR: correct assessment rate; PT: post term; GM: general movement; GM basic: general movements basic training course.

**Table 3 ijerph-18-13428-t003:** The GMA experience of the survey participated assessors and their median CAR in each judgement type.

Type of Assessors		n	Median CAR (%)/IQR									
			All Clips		Normal FMs		F + +		Abnormal FMs		Absent FMs	
All participants		60	55/25		60/27		70/40		50/88		33/34	
GM basic certified	yes	18	65/11	*	67/13	*	90/10	*	50/100	n.s.	67/67	*
	no	42	50/25		53/22		70/30		50/50		33/33	
Practise GMA regularly	yes	21	55/23	n.s.	60/24	n.s.	80/30	n.s.	50/50	n.s.	33/51	n.s.
	no	39	55/30		60/27		70/40		50/50		33/34	
Have watched 3 fidgety GMs videos in the last 3 months	yes	17	60/25	n.s.	60/30	n.s.	80/25	n.s.	50/100	n.s.	33/34	n.s.
	no	43	55/30		60/27		70/40		50/50		33/34	

* *p* < 0.01; n: number; CAR: correct assessment rate; IQR: inter quartile range; GM basic: general movements basic training course; n.s.: not significant.

**Table 4 ijerph-18-13428-t004:** The survey participated assessors and their median CAR in each judgement type.

Survey		n	Median CAR (%)/IQR									
			All Clips		Normal FMs		F + +		Abnormal FMs		Absent FMs	
First ^a^	All clips	28	60/24	*	67/23	*	80/20	n.s.	50/50	n.s.	33/34	*
Follow up	Non-flipped clips	28	84/14		82/18		89/22		100/100		100/33	
Follow up ^b^	Non-flipped clips	22	87/15		82/11		89/22		100/100		100 /33	

^a^ First survey of 28 participants who underwent the follow up survey ^b^ Follow up survey of 22 participants who took CAR 80% more on the flipped clips. * *p* < 0.01; n: number; CAR: correct assessment rate; IQR: inter quartile range; n.s.: not significant.

## Data Availability

The data presented in this study are available on request from the corresponding author.

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
