# Peer review of "Fidgety Movements Assessment Accuracy Survey in Japan"

_ijerph, 2021, doi:10.3390/ijerph182413428_

Round 1

Reviewer 1 Report

This is a fluently written report on an interesting subject. The method and the description of the results is clear and can be repeated. The references are adequate, and the authors are aware of the limitations of the work.

The paper I read reports an investigation of the accuracy of fidgety movements (FMs) assessment. The findings support that course certification is required to practice general movements assessment (GMA). Furthermore, a self-learning exercise with the confirmed judgement FMs clips may be effective for improving the ability of FMs judgement.

As I pointed out the main question addressed by the research is relevant and of interest to doctors. The topic is not original and adds marginally to the subject area compared with other published material. The paper fluently written, and the text is clear and easy to read. The method and the description of the results is clear and can be repeated. The references are adequate. The authors are aware of the limitations of their research works. The conclusions are consistent with the evidence and arguments presented and address the main question posed in the manuscript. I do support publication of this material.

Author Response

Thank you for your review of our paper. It is a great honor to receive your excellent evaluation.

Reviewer 2 Report

All the reported assessments are based on surveys which are very subjective and can be associated with bias. A more objective assessment tool can be used to enhance scientific merit of this manuscript. 

The authors investigated the accuracy of fidgety fetal movements in 18 assessors using a survey based study. I think the topic is interesting as it shows that remote and self learning can improve the accuracy of assessments particularly given the current COVID situation. Although the concept of fidgety movement assessments for potential detection of infants with CP is not totally novel, self learning exercises using video clips would make this study somehow interesting. The text is clear and easy to read. However, I have the following comments: 1) How general movement assessments (GMAs) were exactly conducted? If the values have normal distribution, they should be reported as mean (SD), otherwise as median (IQR). 2) I'd suggest providing a table with a summary of definitions (i.e. what's F++? what's abnormal F?, ...)  3) In the abstract, no data were present regarding how 3-month self learning exercises improved the scores. 4) In the results section, adding a figure like a bar graph or box plot would be helpful. 

Author Response

Thank you for your review of our paper.

Responses to the comments of Reviewer 2

Comments 1.

How general movement assessments (GMAs) were exactly conducted? If the values have normal distribution, they should be reported as mean (SD), otherwise as median (IQR).

Response: As you pointed out, the normal distribution of CARs in abnormal and absent judgement types could not be confirmed. We changed the description of the representative values of CARs in the whole manuscript from the average value to the median value and IQR. We added that to the explanation in Data analysis section. (p4 line 163-164)

Comment 2.

I'd suggest providing a table with a summary of definitions (i.e. what's F++? what's abnormal F?, ...)

Response: As suggested, we added Table 1 as a summary of judgement type and abbreviations (p3 line 100-101). As a result, the Table numbers of all tables have been changed.

Comment 3.

In the abstract, no data were present regarding how 3-month self-learning exercises improved the scores.

Response: As suggested, we have added a data showing the effect of self-learning in the abstract (p1 line 20-21).

Comment 4.

In the results section, adding a figure like a bar graph or box plot would be helpful.

Response: As suggested, we have added two box plots to present significant differences graphically (p6 line 203-204, p7 line 231-232).

Reviewer 3 Report

Thank you for allowing us to review this manuscript. As you pointed out, GMA evaluators are required to have specific, high-quality training, and regular practice and readjustment are essential. I think the results of this study will provide useful information for using GMA.

I would confirm the following issue. Did the author give the correct answer to the participants after completing the initial survey? On line 147, the authors stated, "the assessor thought he or she could assess correctly when the same video was shown as during the first survey" but I don't understand why the assessor feels confident in the second evaluation. Please clarify whether they had been able to know the correct answer.

Author Response

Thank you for your review of our paper. 

Responses to the comments of Reviewer 3

Comment.

Did the author give the correct answer to the participants after completing the initial survey? On line 147, the authors stated, "the assessor thought he or she could assess correctly when the same video was shown as during the first survey" but I don't understand why the assessor feels confident in the second evaluation. Please clarify whether they had been able to know the correct answer.

Response: The participants were given the correct answers at the time of web meeting after the first survey. As suggested, we added the description about this point in the section "2.5. Self-learning exercise after the survey"(p4 line 148-149).

Round 2

Reviewer 2 Report

I can see some improvement in the manuscript and therefore I don't have major objection to accepting this paper for publication. 

Reviewer 3 Report

I have confirmed that the manuscript has been revised well.